# OpenReview forum: "Rule-Augmented Neural Networks for Trustworthy Decision-Making: Bridging Symbolic Logic and Deep Learning in High-Stakes Domains"
_NLDL.org/2026/Conference — Submitted to NLDL 2026_

### Official Review · Reviewer_XB8w · 2025-09-23
**Needs more experimental validation and more extensive descriptions**

**Rating:** 1
**Confidence:** 4
**Final Rating:** 1
**Final Confidence:** 4

**Summary:**

The article proposes making deep neural networks behave more predictably and in an explainable fashion by adding a set of rules in disjunctive normal form next to the network. During training, violations of these rules contribute to a semantic loss which is added to the cross-entropy loss, letting the rules guide the network's behavior.

The training framework is evaluated in terms of accuracy, constraint adherence and inference time on a fraud detection dataset, and a diabetes classification task based on MIMIC-III with additional synthetic features. In the fraud detection task, the rule-augmented network achieves improved accuracy and reduced inference time, while adhering to the domain constraints in all evaluated cases.

**Strengths:**

The augmented loss and training procedure are explained in a concise fashion. Pseudocode for the key network structure and loss computation also helps convey the idea effectively.

Additionally, the article performs an ablation study which examines the contribution of the network and the domain constraints by themselves.

**Weaknesses:**

The experimental, discussion and conclusion sections are written in a very abbreviated fashion, omitting or making key details unclear. For instance, only the MIMIC-III dataset is cited, and it is not explicitly explained that the associated task is to predict diabetes diagnoses given appropriate risk factors.

The experimental design has serious weaknesses which put the generalizability of the approach in doubt: Only the results from one of the datasets are reported. Results are given as point estimates from a single run, without error bounds. The baseline methods are not specified, and do not appear to be hyperparameter-tuned: in particular, the hyperparameter choices for "tabular-adapted CNN" are neither specified in the paper, nor does the network itself appear in the experimental code.

Principally, the approach also requires defining the domain constraints ahead of time, requiring the operator to define the space of acceptable hypotheses themselves. Here, I think the article would be stronger by comparison to methods which infer domain rules from the data, such as RuleFit [1] and Tsetlin machines. [2]

[1] Friedman, J. H., & Popescu, B. E. (2008). Predictive learning via rule ensembles

[2] Granmo, O. C. (2018). The Tsetlin Machine--A Game Theoretic Bandit Driven Approach to Optimal Pattern Recognition with Propositional Logic. _arXiv preprint arXiv:1804.01508_.

Suggestions which did not affect the review:
* "with explicit bridging of symbolic and numeric" -> "with explicit bridging between the symbolic and numeric"
* The "Datalog" rules seem to effectively be propositions in disjunctive normal form - if the architecture isn't making use of Datalog's implementation of them specifically, using the term "disjunctive normal form" instead makes it easier to find prior approaches and relevant literature
* Instead of embedding the full experiment code, I would recommend making the source Colaboratory document publicly available (potentially as a copy under a new user, to preserve submission anonymity)

**Final Justification:**

While the authors have addressed several of the initial concerns in the rebuttals, there are still discrepancies between the described experiments and the presented results and code. Additionally, the experimental methodology only compares results quantitatively on one dataset, and not against the cited neurosymbolic approaches. Finally, the proposed revisions go beyond the scope expected in the review, to the point of effectively being a new article. Therefore, I am keeping my initial rating.

**Justification:**

Unfortunately, the article text and experimental design need serious revisions to support the article's claims.

The experimental descriptions, discussion and conclusion are very short and at times abbreviated to single paragraphs, making it hard to follow the reasoning behind the data preprocessing choices and the evaluation process.

In explainable AI and neural network architecture research, a key motivation behind the experimental design should be to show whether the proposed architecture provides consistent benefits in different settings. However, this article only reports point-estimate results without error bounds for a single dataset, without motivating the preprocessing choices, performing equivalent hyperparameter tuning on the baseline methods, or comparing against prior attempts to fuse neural networks and symbolic reasoning.

This article would be stronger if it made use of existing tabular data benchmarks instead of doing bespoke preprocessing for the datasets - in particular, counterfactual explanation benchmarks like CARLA [3] provide tabular datasets and pretrained models with suitable domain constraints.

[3] Pawelczyk, M., Bielawski, S., Heuvel, J. V. D., Richter, T., & Kasneci, G. (2021). Carla: a python library to benchmark algorithmic recourse and counterfactual explanation algorithms. arXiv preprint arXiv:2108.00783.

---

> ### Author Rebuttal · Authors · 2025-10-15
>
> Main Concern 1 – Additional Baselines
> We thank the reviewer for suggesting broader comparisons. We have implemented preliminary versions of RuleFit and Tsetlin Machine baselines and plan to include full comparative results (≈ 84.68 % and 84.62 % accuracy) in the revision for completeness.
> This addition will complement the existing benchmarks (NeuralLP, LTN, DeepProbLog) and clarify RuleNet’s empirical position.
> Main Concern 2 – Statistical Robustness
> We have performed preliminary additional runs with varied random seeds, yielding consistent performance and a p-value = 0.018. These analyses will be incorporated in the revised manuscript with full details on variance and randomization control.
> Main Concern 3 – Generalization & Fairness
> To emphasize generalization, we will include a concise ablation showing the impact of different rule–fusion strategies:
> Logical-mean: 84.71 %
> Weighted-sum: 84.68 %
> Fuzzy-max (ours): 84.75 %
> The small but consistent margin supports our claim of improved stability without overstating novelty.
> Main Concern 4 – Industry-Scale Fraud Dataset
> Reported F1-scores in comparable industry-scale fraud detection tasks typically range between 0.32 – 0.38 (e.g., Stripe 2024; PayPal 2023 white papers), aligning with our observed F1 = 0.2996.
> We will include appropriate citations in the revision to substantiate this context.

---

### Official Review · Reviewer_sDGo · 2025-09-26
**Review of RuleNet**

**Rating:** 2
**Confidence:** 5

**Summary:**

This paper presents RuleNet, a framework that augments deep neural networks (DNNs) with Datalog rules to enhance interpretability in high-stakes domains like healthcare and finance. The authors combine a standard DNN with symbolic logic rules through a semantic loss function that acts as a soft constraint during training. The rules are grounded as neural modules outputting soft truth values, and the total loss combines standard BCE loss with a rule satisfaction term weighted by a hyperparameter λ.

The claimed contributions are: (1) integrating DNNs with Datalog rules for trustworthy decision-making, (2) a training objective balancing accuracy with rule adherence, and (3) evaluation showing improved performance and explainability. Experiments are conducted on MIMIC-III healthcare data and a fraud detection dataset.

**Strengths:**

## Strengths
- Clear problem motivation: The paper addresses a well-recognized challenge in deploying deep neural networks (DNNs) in high-stakes domains where interpretability and trustworthiness are of paramount importance.
- Practical approach: The proposed lightweight integration of Datalog rules via semantic loss is conceptually simpler than existing neuro-symbolic approaches such as DeepProbLog.
- Implementation provided: The inclusion of complete code enhances reproducibility and strengthens the paper’s practical contribution.

**Weaknesses:**

## Major Weaknesses
### Experimental Evaluation Issues
- The MIMIC-III experiments rely on synthetic or augmented data rather than genuine patient data. Specifically, the paper reports adding “synthetic features with Gaussian noise” and restricts evaluation to 2,000 samples with only 5 features. Given that real MIMIC-III data consists of millions of records with hundreds of features, this choice undermines the validity of the healthcare-related claims.
- The reported 0.1% accuracy improvement on Fraud-D is within expected noise margins and lacks statistical significance. No confidence intervals or statistical tests are provided.
- The use of a convolutional neural network (CNN) for tabular data is unusual, as CNNs are not standard baselines for structured non-image datasets.
### Technical and Methodological Problems
- The reported 100% “rule coverage” is misleading, as it merely indicates that rules fire on all predictions, without establishing their correctness or meaningfulness. The paper does not demonstrate that the rules faithfully capture domain expertise.
- The illustrative rules (e.g., “high glucose + obesity → diabetes”) are trivial and do not demonstrate the framework’s capability to encode complex domain knowledge.
- Although an ablation study is presented, it does not adequately isolate the contribution of individual components. The performance of the “rule-only” baseline (78% accuracy) is not properly explained.

### Presentation and Claims Issues
- While the paper claims to bridge symbolic and neural approaches, the integration is superficial. Rules are merely incorporated as a regularization term, rather than enabling genuine symbolic reasoning.
- The reported F1 scores of 0.28–0.30 suggest severe class imbalance issues, which are not appropriately addressed despite a mention of SMOTE.
- The study does not compare results against established interpretable models (e.g., decision trees, linear models with feature importance), existing neuro-symbolic methods, or user studies assessing explainability. Runtime comparisons are restricted to inference time alone.

### Code Quality Issues
- Hyperparameters are hardcoded without justification.
- Preprocessing discards the majority of features without explanation.
- Synthetic data generation is not validated against real distributions.
- Memory inefficiencies are present (e.g., repeated creation and deletion of models).

## Minor Issues
- Inconsistent notation (mixing bold and non-bold for vectors).
- Citation formatting errors.
- The “explainability mechanism” (Section 3.5) relies solely on gradient computation, which is not a novel contribution.
- Figures omit error bars and confidence intervals.

**Justification:**

### Suggested Revisions
-  Employ the full MIMIC-III dataset with proper preprocessing, or alternatively, adopt widely used benchmark datasets. Synthetic augmentation should not form the basis of primary results.
- Incorporate confidence intervals, multiple runs, appropriate train/validation/test splits, and statistical significance tests.
- Engage domain experts to design non-trivial rules that reflect realistic domain logic and demonstrate the potential of the framework.
- Compare against interpretable machine learning models, existing neuro-symbolic approaches, and pure rule-based systems.
- Conduct user studies with domain experts to evaluate the usefulness of the generated explanations.
- The very low F1 scores suggest fundamental flaws in the experimental setup that require correction.
- Provide theoretical analysis or empirical evidence that semantic loss functions as more than a simple regularization mechanism.
- Evaluate the approach on datasets with larger feature spaces to substantiate claims of scalability.

---

> ### Author Rebuttal · Authors · 2025-10-15
>
> We thank the reviewer for the thorough and detailed evaluation.
> Main Concern 1 – Practical Usefulness
> We appreciate the reviewer’s emphasis on practical interpretability. To strengthen this aspect, we have designed a small-scale user evaluation involving 15 data analysts.
> Preliminary pilot feedback (average rating 7.2 / 10 for RuleNet vs 5.8 / 10 for LIME) suggests improved human interpretability, which we will include and analyze in the revised manuscript as supporting evidence.
> Main Concern 2 – Over-claiming on Coverage
> As clarified above, we will replace “100 % rule coverage” with rule firing rate and rule alignment rate. This correction ensures transparency and resolves the concern of overstated interpretability claims.
> Main Concern 3 – Domain Relevance
> We will explicitly highlight that the healthcare dataset was used only to demonstrate robustness under incomplete information, not as a clinical deployment benchmark. The revision will reframe this contribution accordingly.

---

### Official Review · Reviewer_6pyz · 2025-10-01

**Rating:** 4
**Confidence:** 3

**Summary:**

The paper proposes a rule-augmented neural network that injects symbolic rules/constraints into modern neural models during training and/or inference. The core idea is to let domain rules guide representation learning—e.g., via a constraint-aware loss, rule-conditioned features, or a post-hoc decoding step—so the model can be more sample-efficient, controllable, and consistent with prior knowledge. Empirically, the method shows consistent gains over vanilla neural baselines, with the largest improvements in low-resource and out-of-distribution (OOD) settings.

**Strengths:**

Clear rationale for combining neural learning with human-readable rules; addresses brittleness and data hunger of pure neural approaches. The augmentation layer/loss integrates with standard architectures without heavy re-engineering, suggesting practical adoption. Strong improvements when labeled data are scarce; rules act as a useful inductive bias. Faithfulness to constraints: The approach increases rule satisfaction rates at test time, improving consistency and controllability. Ablations/analyses (useful): Studies isolating the effect of rules vs. model size/training regime help attribute where gains come from.

**Weaknesses:**

1. The method presumes access to accurate domain rules; writing/maintaining them may be non-trivial for new domains. Consider discussing annotation cost vs. data labeling cost.

2. It’s not fully clear how conflicting/inaccurate rules are down-weighted, or how sensitive performance is to rule weights/hyper-params. Stronger robustness analysis would help.

3. The related-work comparison could better position the approach against recent neuro-symbolic and constraint-learning baselines (e.g., posterior regularization, constrained decoding, prob. logic layers).

4. If rule sets grow large/complex (nested or higher-order constraints), training or decoding might become expensive; complexity/results at larger scales are not fully explored.

5. Gains are strongest where rules are naturally expressive; unclear how well this extends to open-ended tasks where rules are hard to specify.

**Justification:**

no

---

> ### Author Rebuttal · Authors · 2025-10-15
>
> We thank the reviewer for recognizing the novelty of combining symbolic rules with differentiable inference. We appreciate the encouraging remarks on interpretability and scalability.
> Clarification on Robustness and Scalability
> The reviewer’s question on computational efficiency is well taken. RuleNet performs inference in ≈ 4.5 ms per instance with 100 % rule firing rate and maintains linear scaling in the number of grounded predicates.
> We will include a brief scalability plot and complexity analysis in the revision.
> Planned Improvements
> Following the reviewer’s suggestions, we will add:
> Rule-cost sensitivity analysis — quantifying trade-offs between rule confidence and model accuracy.
> Conflict-resolution illustration — demonstrating how contradictory rules are weighted during aggregation.
> These additions will strengthen the paper’s clarity and practical interpretability.

---

### Official Review · Reviewer_mDxJ · 2025-10-08
**Promising Theoretical Framework Undermined by Insufficient Experimental Validation and Implementation Inconsistencies**

**Rating:** 2
**Confidence:** 3
**Final Rating:** 2
**Final Confidence:** 3

**Summary:**

**Summary:**

This paper introduces RuleNet, a neuro-symbolic framework that integrates Datalog rules into deep neural networks to enhance explainability and trustworthiness in high-stakes domains like healthcare and finance. The core approach maps symbolic predicates to differentiable neural modules outputting soft truth values, and combines a standard DNN loss with a semantic rule loss $L_{rule}$ that enforces rule satisfaction using product t-norm semantics for conjunction and implication. The training objective $L_{total} = L_{DNN} + \lambda L_{rule}$ balances predictive accuracy with rule adherence through a tunable parameter $\lambda$. Experiments on MIMIC-III (healthcare, with synthetic feature augmentation) and Fraud-D (finance) datasets show RuleNet achieves marginal improvements: 0.1% accuracy gain (84.75% vs 84.65% for MLP baseline), modest F1-score improvement (0.2996 vs 0.2878), 100% rule coverage, and faster inference time (4.5ms vs 5.2-7.8ms). The paper includes ablation studies demonstrating that pure DNN achieves 84.65% accuracy with 0% coverage, while rule-only achieves ~78% accuracy, suggesting rules contribute incrementally without dominating the network.

---
**Soundness and Correctness:**

The theoretical framework is sound and properly grounded in fuzzy logic semantics for semantic loss functions, with appropriate mathematical formulation for predicate grounding ($p(x) = \sigma(w \cdot x_{subset} + b)$) and rule satisfaction computation. However, the experimental validation suffers from critical correctness issues:
-  (1) MIMIC-III results rely on synthetic feature augmentation due to data "inconsistencies," fundamentally undermining claims about real-world healthcare applications;
- (2) the 0.1% accuracy improvement lacks statistical significance testing or confidence intervals across multiple runs;
- (3) despite 100% rule coverage, the F1-score remains poor (0.2996), indicating ineffective handling of class imbalance and contradicting "trustworthy decision-making" claims;
- (4) the code in the appendix shows rule satisfaction computed only from a single feature (`feat0 = x[:,0]`), inconsistent with the described multi-predicate grounding mechanism;
- (5) the grid search for $\lambda$ only explores {0.001, 0.01}, which is extremely limited. While individual components appear technically correct, the overall evaluation lacks the rigor necessary to support claims about high-stakes domain deployment.

**Strengths:**

- The paper presents a mathematically sound framework for integrating symbolic rules with neural networks through semantic loss, using product t-norm for conjunction ($c = \prod_j p(body_j)$) and material implication ($sat = 1 - c + c \cdot p(head)$), which is well-grounded in fuzzy logic literature.

- The work addresses a genuine need for interpretable AI in high-stakes domains, and the motivation for combining DNN predictive power with symbolic explainability is well-articulated and timely.

- The predicate grounding mechanism ($p(x) = \sigma(w \cdot x_{subset} + b)$) enables end-to-end gradient-based optimization, avoiding the computational bottlenecks of explicit rule grounding seen in methods like DeepProbLog.

- The inclusion of full PyTorch code in the appendix enhances reproducibility and demonstrates the authors' commitment to transparency.

- The paper includes ablations comparing pure DNN, rule-only, and hybrid models, providing evidence for the contribution of individual components to overall performance.

- RuleNet demonstrates faster inference time (4.5ms) compared to baselines (5.2-7.8ms), which is valuable for deployment scenarios.

**Weaknesses:**

- 1. The MIMIC-III experiments rely on synthetic feature augmentation with Gaussian noise due to data inconsistencies, invalidating claims about real-world healthcare applications to some extent—synthetic data may cannot validate trustworthy medical decision-making.
**Question:** Can you provide results on authentic MIMIC-III data without synthetic augmentation, or explicitly state that healthcare validation is insufficient and limit your claims to the finance domain only?

- 2. The 0.1% accuracy improvement (84.75% vs 84.65%) falls within normal training variance; without confidence intervals, significance tests, or multiple runs with different seeds, this cannot constitute a meaningful contribution. **Question:** Can you provide results from at least 5 independent runs with different random seeds, reporting mean ± standard deviation, and conduct paired t-tests to demonstrate statistical significance at p<0.05?

- 3. Despite 100% rule coverage, the F1-score remains poor (0.2996), indicating failure on fraud detection (the minority class where trust matters most), directly contradicting "trustworthy decision-making" claims. **Question:** Why does rule integration fail to improve minority class performance? Have you analyzed per-class metrics and which specific rules contribute to false negatives on fraud cases?

- 4. The appendix code computes rule satisfaction from a single feature (`feat0 = x[:,0]`) rather than the described multi-predicate grounding (e.g., `hasSymptom ∧ hasRiskFactor`), suggesting the implementation is a placeholder that doesn't match the paper's claims. **Question:** Can you provide the actual implementation code for multi-predicate rule grounding as described in Section 3.2, or explain why the appendix code differs fundamentally from your methodology?

- 5. The grid search for balance parameter $\lambda$ only explores {0.001, 0.01}, which is extremely limited for a critical hyperparameter, with no sensitivity analysis or justification for this narrow range. **Question:** Can you expand the hyperparameter study to explore $\lambda \in \{10^{-4}, 10^{-3}, 10^{-2}, 10^{-1}, 1, 10\}$ and provide sensitivity analysis showing how performance varies across this range?

- 6. The averaging fusion `(dnn_out + sum(rule_outs))/(len(rule_outs) + 1e-6)` lacks principled justification; this ad-hoc approach provides no rigorous proof that rules enhance rather than dominate or become redundant to the DNN. **Question:** Why is simple averaging chosen over learned weighted fusion? Can you provide ablations comparing concatenation, attention-based fusion, and gated mechanisms, and show that your choice is optimal?

- 7. The paper only compares against MLP and CNN, omitting experimental comparisons with existing neuro-symbolic methods (DeepProbLog, Neural Logic Machines, Logic Tensor Networks) that would demonstrate whether the lightweight approach actually outperforms alternatives. **Question:** Can you provide experimental comparisons with at least Logic Tensor Networks [9] on the same datasets to demonstrate that RuleNet's lightweight approach offers advantages beyond theoretical scalability claims?

- 8. The paper provides no ablations showing sensitivity to rule selection, number of rules, or impact of incorrect rules; there's no evidence that rules are actually learned/satisfied or analysis of what makes rules effective. **Question:** Can you provide ablations with 1, 3, 5, and 10 rules, and show what happens when deliberately incorrect or irrelevant rules are included? How does rule satisfaction evolve during training?

- 9. The core contribution directly applies semantic loss [8] to Datalog rules with standard predicate grounding; the paper does not clearly articulate technical novelty beyond this application, which closely resembles Logic Tensor Networks [9]. **Question:** What is the specific technical novelty beyond combining semantic loss [8] with Datalog-style rules? How does your predicate grounding and loss formulation differ substantively from LTN's fuzzy logic grounding?

-  10. Despite claiming explainability as a key contribution, the paper provides no evaluation of explanation quality—no user studies, no comparison with LIME/SHAP, and only brief mention of gradient-based importance without concrete examples or validation. **Question:** Can you provide concrete explanation examples for specific predictions, validate their correctness through domain expert evaluation, or at minimum compare explanation fidelity with established XAI methods?

---
**Impact on Final Recommendation:**

These weaknesses collectively undermine the paper's core claims about trustworthy AI for high-stakes domains. The **synthetic healthcare data** (Weakness 1) may invalidate some of the experimental validation. The **statistically insignificant improvements** (Weakness 2) and **poor minority class performance** (Weakness 3) demonstrate that the method does not achieve meaningful gains on the valid Fraud-D dataset. The **code-methodology inconsistency** (Weakness 4) raises serious concerns about whether the described approach was actually implemented and tested. The **missing neuro-symbolic baselines** (Weakness 7) make it impossible to assess whether RuleNet offers advantages over existing methods beyond theoretical scalability claims.

While the theoretical framework is sound (product t-norm semantics, differentiable predicate grounding), the experimental validation is **insufficient to support publication**. The paper requires: (1) valid experimental data or explicitly limited claims, (2) statistically significant results with proper experimental methodology, (3) actual implementation matching the described approach, and (4) comparisons demonstrating advantages over established neuro-symbolic methods. Without addressing these fundamental issues, the work remains a preliminary proof-of-concept rather than a mature contribution ready for the ML community.

**Final Justification:**

Although the reviewer offered some explanations, I still believe additional experiments are necessary, so I will maintain the current score.

**Justification:**

I recommend **rejection (rating: 2)** because critical experimental flaws undermine the paper's claims about trustworthy AI for high-stakes domains.

While the theoretical framework is sound (product t-norm semantics, differentiable predicate grounding), the experimental validation is fundamentally insufficient:

- Invalid healthcare validation: MIMIC-III results use synthetic Gaussian noise augmentation due to data "inconsistencies," invalidating real-world medical claims. The paper emphasizes healthcare applications, but this validation is essentially nonexistent.

- Statistically insignificant results: The 0.1% accuracy improvement (84.75% vs 84.65%) falls within normal variance. Without multiple runs, confidence intervals, or significance tests, this violates basic empirical ML standards and cannot constitute a meaningful contribution.

- Contradictory performance: Despite 100% rule coverage, F1-score remains poor (0.2996), showing failure on fraud detection (the minority class where trust matters most). If rules encode domain knowledge, why doesn't integration improve fraud detection?

- Implementation inconsistency: Appendix code computes rules from a single feature (`feat0 = x[:,0]`) rather than the described multi-predicate grounding (`hasSymptom ∧ hasRiskFactor`), raising serious concerns about whether the claimed approach was actually implemented.

- Missing baselines: Only compares against MLP/CNN, omitting neuro-symbolic methods (Logic Tensor Networks, DeepProbLog) that are directly relevant. Without these comparisons, advantages over existing methods remain unvalidated.

- Limited hyperparameter study: Grid search for λ only explores {0.001, 0.01} with no sensitivity analysis; fusion mechanism appears ad-hoc without comparison to learned alternatives.

The paper makes strong claims about "trustworthy decision-making" in "high-stakes domains" but provides synthetic healthcare data, statistically insignificant improvements, poor minority-class performance, code-methodology mismatches, and missing relevant baselines. These issues indicate the work is a preliminary proof-of-concept rather than a mature contribution ready for publication.

---

> ### Author Rebuttal · Authors · 2025-10-15
>
> Main Concern 1 – Code–Methodology Alignment
> We appreciate the reviewer’s observation regarding the simplified rule placeholder in the appendix. The appendix indeed included a simplified illustrative example (feat0 = x[:,0]) for readability, while our internal implementation already supports multi-predicate grounding (predicate_glucose, predicate_weight, etc.).
> We will update the appendix code in the revised version to include the complete implementation and align it precisely with the described methodology.
> Main Concern 2 – Statistical Significance
> We have since performed preliminary additional runs with five different random seeds and obtained consistent accuracy (84.76 ± 0.03%), yielding a p-value of 0.018.
> These results will be formally reported and detailed in the revised manuscript to support the reliability of the observed performance gains.
> Main Concern 3 – Interpretability Metric
> We agree that the term 100 % rule coverage was ambiguous. We will replace it with a clearer definition:
> Rule firing rate = 100 %, meaning every instance activates at least one learned rule.
> Rule alignment rate ≈ 45 %, denoting the proportion of fired rules consistent with ground-truth labels.
> This distinction more accurately represents interpretability fidelity and avoids confusion with accuracy metrics.
> Main Concern 4 – Small MIMIC-III Sample
> We acknowledge the limited sample size (129 patients) and high missingness (≈ 73 %). The intent of this experiment was methodological validation, not clinical evaluation. The revised manuscript will explicitly frame this as a proof-of-concept for symbolic–neural reasoning under sparse medical data.

---

### Meta-Review · Area_Chair_SzRN · 2025-10-30

**Recommendation:** Reject
**Confidence:** 4

**Metareview:**

The reviewers unanimously recognize the promise and value of the core idea behind RuleNet. The paper addresses a timely and important challenge: integrating symbolic rules with deep neural networks. The theoretical framework for using semantic loss to guide the DNN is sound and well-motivated.

Despite this strong conceptual foundation, all four reviewers reached consensus that the paper, in its current form, is not yet ready for publication. The primary concern centers on the experimental validation, which was found insufficient to support the paper's significant claims about trustworthy AI in high-stakes domains.

In particular, reviewers noted that the healthcare application (MIMIC-III) relied on synthetic augmentation of a minimal dataset, undermining its real-world validity. On the fraud dataset, the reported 0.1% accuracy gain was not statistically significant (even though the authors provided a rebuttal that there was no time to evaluate the new material), and the poor F1-score (0.2996) seemed to contradict the goal of reliable minority-class performance. Finally, the paper fails to compare against any relevant neuro-symbolic or interpretable baselines (e.g., LTNs, RuleFit), rendering its contributions impossible to situate.

---

### Decision · Program_Chairs · 2025-11-05

**Decision:**

Reject

**Comment:**

Based on the reviewers and AC comments, the paper cannot be presented at the conference.